# Multicenter Observational Study on Metastatic Non-Small Cell Lung Cancer Harboring *BRAF* Mutations: Focus on Clinical Characteristics and Treatment Outcome of V600E and Non-V600E Subgroups

**DOI:** 10.3390/cancers14082019

**Published:** 2022-04-16

**Authors:** Fabiana Perrone, Giulia Mazzaschi, Roberta Minari, Michela Verzè, Cinzia Azzoni, Lorena Bottarelli, Rita Nizzoli, Monica Pluchino, Annalisa Altimari, Elisa Gruppioni, Francesca Sperandi, Elisa Andrini, Giorgia Guaitoli, Federica Bertolini, Fausto Barbieri, Stefania Bettelli, Lucia Longo, Maria Pagano, Candida Bonelli, Elena Tagliavini, Davide Nicoli, Alessandro Ubiali, Adriano Zangrandi, Serena Trubini, Manuela Proietto, Letizia Gnetti, Marcello Tiseo

**Affiliations:** 1Medical Oncology Unit, University Hospital of Parma, 43126 Parma, Italy; fabiana.perrone89@libero.it (F.P.); giulia.mazzaschi@unipr.it (G.M.); mverze@ao.pr.it (M.V.); rnizzoli@ao.pr.it (R.N.); mpluchino@ao.pr.it (M.P.); mtiseo@ao.pr.it (M.T.); 2Department of Medicine and Surgery, University of Parma, 43126 Parma, Italy; 3Unit of Pathological Anatomy, University Hospital of Parma, 43126 Parma, Italy; azzoni@ao.pr.it (C.A.); lorena.bottarelli@unipr.it (L.B.); lgnetti@ao.pr.it (L.G.); 4Pathology Unit, IRCCS Azienda Ospedaliero-Universitaria di Bologna, 40138 Bologna, Italy; annalisa.altimari@aosp.bo.it (A.A.); elisa.gruppioni@aosp.bo.it (E.G.); 5Medical Oncology, IRCCS Azienda Ospedaliero-Universitaria di Bologna, 40138 Bologna, Italy; francesca.sperandi@aosp.bo.it; 6Department of Experimental Diagnostic and Specialized Medicine (DIMES), Alma Mater Studiorum, University of Bologna, 40126 Bologna, Italy; elisa.andrini2@studio.unibo.it; 7Division of Medical Oncology, University Hospital of Modena, 41125 Modena, Italy; gioguaitoli@gmail.com (G.G.); bertolini.federica@policlinico.mo.it (F.B.); faustobarbieri@tin.it (F.B.); 8Ph.D. Program Clinical and Experimental Medicine (CEM), Department of Biomedical, Metabolic and Neural Sciences, University of Modena and Reggio Emilia, 41125 Modena, Italy; 9Pathology Unit, University Hospital of Modena, 41125 Modena, Italy; bettelli.stefania@policlinico.mo.it; 10Medical Oncology Unit, Sassuolo Hospital, AUSL Modena, 41121 Modena, Italy; l.longo@ausl.mo.it; 11Medical Oncology Unit, Clinical Cancer Centre, Azienda USL-IRCCS Reggio Emilia, 42122 Reggio Emilia, Italy; maria.pagano@ausl.re.it (M.P.); candida.bonelli@ausl.re.it (C.B.); 12Pathology Unit, Clinical Cancer Centre, Azienda USL-IRCCS Reggio Emilia, 42122 Reggio Emilia, Italy; elena.tagliavini@ausl.re.it; 13Molecular Biology, Oncology and Advanced Technology Unit, Azienda USL-IRCCS Reggio Emilia, 42122 Reggio Emilia, Italy; davide.nicoli@ausl.re.it; 14Pathology Unit, AUSL Piacenza, 29121 Piacenza, Italy; a.ubiali@ausl.pc.it (A.U.); a.zangrandi@ausl.pc.it (A.Z.); s.trubini@ausl.pc.it (S.T.); 15Medical Oncology Unit, AUSL Piacenza, 29121 Piacenza, Italy; m.proietto@ausl.pc.it

**Keywords:** non-small-cell lung cancer (NSCLC), BRAF, V600E, real-life, target therapy

## Abstract

**Simple Summary:**

Around 2–4% of lung adenocarcinoma harbors BRAF mutations. Dabrafenib and Trametinib represent the first treatment-choice for BRAF V600E^mut^ NSCLC, regardless of the line of therapy, while non-V600E^mut^ receive standard immunotherapy or chemo-immunotherapy. Our real-life multicenter study on 44 BRAF mutant NSCLC responds to the urgent need to characterize this subset of patients in-depth, potentially offering new valuable biological and clinical insights. We specifically focused on similarities/discrepancies between V600E and non-V600E populations, providing consistent data about clinicopathologic characteristics, treatment response, and survival outcome.

**Abstract:**

Introduction: BRAF mutation involved 2–4% of lung adenocarcinoma. Differences in clinicopathologic features and patient outcome exist between V600E and non-V600E BRAF mutated NSCLC. Thus, we sought to assess the frequency and clinical relevance of BRAF mutations in a real-life population of advanced-NSCLC, investigating the potential prognostic significance of distinct genetic alterations. Materials and Methods: The present multicenter Italian retrospective study involved advanced BRAF mutant NSCLC. Complete clinicopathologic data were evaluated for BRAF V600E and non-V600E patients. Results: A total of 44 BRAF^mut^ NSCLC patients were included (V600E, *n* = 23; non-V600E, *n* = 21). No significant differences in survival outcome and treatment response were documented, according to V600E vs. non-V600E mutations, although a trend towards prolonged PFS was observed in the V600E subgroup (median PFS = 11.3 vs. 6.0 months in non-V600E). In the overall population, ECOG PS and age significantly impacted on OS, while bone lesions were associated with shorter PFS. Compared to immunotherapy, first-line chemotherapy was associated with longer OS in the overall population, and especially in the BRAF V600E subtype. Conclusions: Here, we report on real-life data from a retrospective cohort of advanced-NSCLC harboring BRAF alterations. Our study offers relevant clues on survival outcome, therapeutic response, and clinicopathologic correlations of BRAF-mutant NSCLC.

## 1. Introduction

*V-Raf murine sarcoma viral oncogene homolog (BRAF)* mutations have been described in 8% of cancer, with higher incidence reported in melanoma (40–60%), papillary thyroid carcinomas (30–70%), and colorectal cancers (5–20%) [1,2]. The *BRAF* gene was localized on chromosome 7 and encodes for the BRAF protein, a RAF kinase which plays a key role in the MAPK cascade, a signaling pathway involved in cell growth, differentiation, proliferation, senescence, and apoptosis [3,4,5,6]. The most common activating *BRAF* mutation found in cancer is a point mutation in exon 15 (c.1799T > A) that corresponds to a valine to glutamate substitution at codon 600 (V600E) [7]. The mutation leads to a 500-fold increase in the kinase activity of BRAF and allows BRAF to be active, regardless of RAS-mediated activation [8,9]. In lung cancer, *BRAF* is mutated in approximately 2–4% of cases [10]. Different from melanoma, in which the vast majority of *BRAF* mutations occur at the V600 site, only approximately 50% of *BRAF* mutant lung adenocarcinomas harbor V600 mutations, with the rest of the cases carrying non-V600 mutations in exons 11 and 15 [1,10]. Compared to other oncogenic drivers, *BRAF* mutations are more frequent in current/former smokers, less frequently mutually exclusive, and associated with higher PD-L1 expression and better response to immunotherapy [11,12,13,14]. The prognostic role of *BRAF* V600E mutation in lung cancer is still unclear. However, multiple evidence reported poorer outcomes and lower response rate to platinum-based chemotherapy in *BRAF*-mutated NSCLC patients compared to *BRAF-*wild type [15]. The therapeutic landscape of *BRAF* V600 mutant NSCLC has been dramatically revolutionized in the last few years. Indeed, based on the encouraging results achieved in melanoma, several studies have investigated the association between BRAF and MEK inhibitors in *BRAF* mutant NSCLC. Additionally, as a consequence of the favorable results of a phase II, single-arm study in both pre-treated and treatment-naïve NSCLC patients harboring *BRAF* V600E mutations, the combination of dabrafenib plus trametinib, has become the standard of care for these patients [16,17,18]. On the other hand, chemotherapy and immunotherapy represent the treatment of choice in non-V600E mutant NSCLC, given the undefined efficacy of BRAF and MEK inhibitor against this molecular alteration [19]. However, in light of the association of some non-V600E mutations, such as G469A and L597R, with high BRAF kinase activity, dual BRAF ± MEK blockage could be conceivably employed, also in this setting [10,20]. Several previous studies have investigated the incidence, distribution, and prognosis of *BRAF* mutant lung adenocarcinoma, but, partly due to relatively small sample size and the retrospective nature, the results have been demonstrated as inconclusive and/or contradictory [11,12,21,22].

Here, we present real-world data on a large cohort of advanced NSCLC patients harboring *BRAF* mutations from five Italian Institutions in the attempt to describe clinical features and survival outcomes, with particular attention given to potential differences between *BRAF* V600E mutated and non-V600E mutated subgroups.

## 2. Materials and Methods

### 2.1. Patients and Study Objectives

The present multicenter retrospective observational study (DETECTION_KB study) was carried out in five Italian Centers (Piacenza, Parma, Reggio Emilia, Modena, and Bologna), involving Oncology and Pathology Units, and included consecutive patients with advanced NSCLC with a *BRAF* mutation diagnosed between 1 January 2018 and 28 February 2020. Patients had to be >18 years old and have data available for clinical follow-up.

The primary aim of the study was to assess the incidence of *BRAF* mutation in a real-life cohort of advanced-NSCLC patients and to illustrate the different type of detected mutations. As exploratory aims, we explored the impact of distinct *BRAF* mutations (V600E vs. non-V600E) on survival outcome and treatment response and the potential association with specific clinicopathological characteristics.

### 2.2. Study Design and Data Collection

Participating centers entered data into a database and completed a survey on the molecular procedures for the diagnostic approach in testing *BRAF* mutation (see Appendix B). The BRAF molecular testing used to analyze the tissue biopsy or cytological specimens had to have similar sensitivity in all participating centers. The following data were extracted from medical records: patient characteristics (date of diagnosis, age at diagnosis, sex, Eastern Cooperative Oncology Group performance status, smoking status); disease history (disease stage at diagnosis, tumor histology, presence of *BRAF* mutations, site of diagnostic biopsy). Outcome data for analysis were also collected from medical records, including the start and end dates of systemic therapy, number of lines of treatment, type of response (complete response, partial response, stable disease, and progression of disease (CR, PR, SD and PD)), date of radiologic disease progression (according to the RECIST criteria version 1.1), date of death, and survival outcomes (PFS, progression-free survival and OS, overall survival). The study was conducted in accordance with the current revisions of the Declaration of Helsinki (Fortaleza, Brazil, 2013) and Italian laws on observational studies; ethical approval was obtained for all five Centers. Written informed consent for trial participation was obtained from all patients (expect for deceased patients).

### 2.3. Statistical Analysis

Anonymous data were collected by each participating Center and treated according to Italian national laws and General Data Protection Regulations. Data were subsequently analyzed by the Medical Oncology Unit of Parma University Hospital.

Comparative analyses were performed for patients with *BRAF* V600E versus non-V600E mutations, and to assess differences between groups, Fisher’s exact test and the Mann-Whitney test were employed for categorical variables and continuous variables, respectively.

OS and PFS were estimated according to the Kaplan Meier method. OS was defined as the interval from the date of diagnosis of metastatic disease to the date of death or last follow-up for alive patients. PFS was defined as the interval from the date of diagnosis of metastatic disease to the date of radiological/clinical progression or death due to any cause, whichever occurred first, or last follow-up visit for patients alive without disease progression. Cut-off for survival analysis was set at 30 March 2021. Median follow-up was calculated according to the so-termed ‘reverse Kaplan Meier’ (Kaplan Meier estimate of potential follow-up) technique [23].

Log-rank test (Mantel Cox) was applied to evaluate statistical differences in PFS and OS between subgroups. Univariate and multivariate Cox proportional hazards regression models were used to test potential prognostic factors both in terms of PFS and OS, and results were presented with hazard ratios (HRs), 95% CI, and *p* values.

Complete and partial responses (CR and PR, respectively) were gathered, with stable diseases (SD) lasting at least 6 months in the disease control group (DC = CR + PR + SD ≥ 6 months), set against the progressive disease (PD) group, which comprises PD and SD lasting less than 6 months.

For all the analyses described above, a *p* value of 0.05 was set as a threshold of statistical significance. To minimize the risk of multiplicity, Bonferroni correction test was applied to all our multiple comparisons. All statistical analyses were carried out using the Statistical Package for the Social Sciences program (SPSS), version 25.0, (IBM, Armonk, NY, USA).

## 3. Results

### 3.1. Baseline Patient Characteristics

From 1 January 2016 to 28 February 2020, an overall number of 2342 patients with advanced NSCLC were tested for *BRAF* mutation, as per clinical practice, by the five Pathology Units involved in the DETECTION_KB study. Of all patients, 127 (5.4%) were *BRAF* mutated: 68 (2.9%) carried a V600E mutation and 59 (2.5%) another (non-V600E) mutation in the *BRAF* gene (Consort Diagram—Figure 1).

At data cut-off, 44 NSCLC cases with available clinical data and adequate follow-up were included, of which 23 presented *BRAF* V600E mutation (V600E group, 52.2%) and 21 (47.8%) displayed *BRAF* mutations other than V600E (non-V600E). Out of 21 non-V600E NSCLC, a considerable proportion (38.1%, *n* = 8) harbored G469A/V/S mutations and an equal fraction carried G466A (*n* = 5, 23.8%) and D594G/N (*n* = 5, 23.8%) alterations. Two cases (9.5%) displayed *BRAF* G464G mutation, while only one had Q257K mutation (Consort Diagram—Figure 1).

Baseline clinicopathologic characteristics of our patient population are reported in Table 1. Median age was 67 years, ranging from 47 to 84, and male patients accounted for 64% of the overall population. The vast majority of patients had 0 or 1 performance status (89%), and 27 (61%) cases presented with more than three metastatic sites at the time of diagnosis. Concerning the histological subtypes, 36 patients had adenocarcinoma (ADC) histology (82%), while 16% and 2% presented not otherwise specified (NOS) NSCLC and squamous cell carcinoma (SCC), respectively. Whitin tested cases (71%), PD-L1 Tumor Proportional Score (TPS) resulted in ≥50% in 16 patients (37%), while the proportion of PD-L1 negative (TPS < 1%) and intermediate (TPS 1–49%) samples was essentially comparable.

Chemotherapy-based regimens represented the most employed first-line treatment (68%), while 21% and 11% patients were administered with single-agent immunotherapy and BRAF/MEK inhibitors, respectively (four cases underwent Dabrafenib plus Trametinib and one patient Encorafenib plus Binimetinib).

Concerning the therapeutic course, among 22 patients who started a second-line therapy, 9 (41%) received single-agent immunotherapy, 6 (27.3%) chemotherapy, and only 1 (4.5%) chemo-immunotherapy combination. Six *BRAF* V600E mutant NSCLCs were treated with anti-BRAF/MEK combination of Dabrafenib and Trametinib. The proportion of cases undergoing third- and fourth-line was 18.2% (*n* = 8) and 2.3% (*n* = 1), with an equal distribution among chemotherapy- and immunotherapy-treated groups.

With a median follow-up of 36.1 months (95% CI 17.9–54.3), median OS and PFS were 13.9 (95% CI 8.4–19.6) and 6.2 (95% CI 5.6–6.9), respectively (Figure 2A*i*,A*ii*). In terms of response to first-line therapy, 24 patients (64.9% of evaluable patients) experienced DC, while 13 (35.1%) experienced PD; 7 were not evaluable.

### 3.2. BRAF Mutations and Baseline Clinicopathologic Characteristics

We subsequently evaluated whether the distribution of clinicopathologic features differed according to specific *BRAF* mutations (V600E vs. non-V600E). As reported in Table 1, no significant variations in terms of number or location of metastatic sites, gender, performance status, age, and first-line treatment were detected between V600E and non-V600E subgroups. Intriguingly, a strong predominance of non-smokers was present in V600E subset (27% vs. 0% in non-V600E, *p* = 0.032), and NSCLC NOS appeared to be less represented in the non-V600E subgroup (22% vs. 9%, *p* = 0.024). Moreover, when evaluating PD-L1 status, *BRAF* V600E samples displayed a trend towards lower levels of PD-L1 expression, with a relatively higher proportion of TPS < 1% (39% vs. 29% in non-V600E, not significant (NS)). Nonetheless, considering the relatively limited sample size and the high number of comparisons, when the obtained statistically significant *p* values were adjusted following Bonferroni’s correction test, no meaningful differences were observed (Appendix A).

### 3.3. BRAF Mutational Status and Impact on Clinical Outcome

Partly as a consequence of the relatively limited sample size, *BRAF* mutational status did not display a striking impact on OS, with a median OS of 13.3 months (95% CI 10.3–16.3) for *BRAF* V600E and 15.1 months (95% CI 4.4–29.7) for *BRAF* non-V600E (Figure 2B*i*). Nevertheless, a tendency towards prolonged PFS was apparent in patients carrying *BRAF* V600E mutation, with a median PFS of 11.3 months (95% CI 3.3–19.2) compared to 6.0 months (95% CI 5.4–6.6) in *BRAF* non-V600E (Figure 2B*ii*). When focusing on disease response to first-line treatment, the proportion of patients experiencing DC and PD was largely comparable between the two *BRAF* subgroups (Figure 2C).

### 3.4. Impact of Clinicopathologic Characteristics on Clinical Outcome

In order to provide useful information about the prognostic and predictive value of widely recognized clinicopathologic characteristics, we assessed the impact of baseline clinical and histopathologic features on patient outcome in the overall population and in *BRAF* V600E vs. non-V600E subgroups. In terms of OS, only ECOG PS and age appeared to significantly condition patient outcome (Table 2). Specifically, patients with ECOG 0–1 and age below the median (<67 years) displayed prolonged OS compared to their counterpart (Cox regression univariate analysis, *p* < 0.05). Although without reaching statistical significance, a trend towards favorable prognosis was documented in female patients (*p* = 0.110, HR = 0.51) and without liver metastatic involvement (*p* = 0.095, HR = 2.26).

With regards to PFS, while most of clinicopathologic parameters did not significantly affect the outcome, the evidence of bone lesions was meaningfully associated with a shorter PFS (*p* = 0.020) (Table 2).

When challenged on multivariate analysis, only ECOG PS maintained their statistically significant impact on OS (data not shown).

We next explored the prognostic value of the abovementioned clinicopathologic characteristics within the two *BRAF* mutational subgroups, thus revealing slight differences. In terms of OS, ECOG PS and age confirmed their prognostic value only in the *BRAF* non-V600E subgroup, while histotype (NSCLC NOS vs. ADC/SCC, *p* = 0.04) and presence of brain metastases (*p* = 0.032) negatively conditioned OS of *BRAF* V600E cases. When evaluating PFS, male sex and lymph nodal involvement resulted significant only in *BRAF* non-V600E, while in *BRAF* V600E subgroup, we did not identify any prognostically relevant factor (Table 2).

Even following Bonferroni’s correction test, the statistical significance of our results was maintained (Appendix A).

### 3.5. Impact of First-Line Therapeutic Strategy on Clinical Outcome

In the perspective of patient management optimization, we finally sought to investigate the putative prognostic role of chemotherapy- vs. immunotherapy-based regimens in our cohort of *BRAF* mutant NSCLC, excluding those patients treated with first-line targeted agents. As shown in Figure 3A*i*, in the overall population, the positive impact of CT-based treatment was remarkable in terms of OS (Median OS: 17.6 vs. 7.1 mos. in IT-treated patients, *p* = 0.074), although not translated in an analogous PFS advantage (Figure 3A*ii*). Conversely, the survival outcome of IT-treated patients was disappointing, registering median OS and PFS of 7.1 and 6.2 months, respectively.

When stratifying patients in *BRAF* V600E and non-V600E subgroups, we demonstrated that *BRAF* V600E mutant NSCLC treated with first-line chemotherapy displayed a significantly prolonged OS compared to those receiving single-agent immunotherapy (Median OS = 20.4 vs. 7.0 mos. in IT-treated patients, *p* = 0.019; Figure 3B*i*), while any difference was observed in *BRAF* non-V600E subset (Figure 3C*i*). In both *BRAF* V600E and non-V600E patients, PFS was not meaningfully affected (Figure 3B*ii*,C*ii*).

Finally, we focused on *BRAF* V600E mutant NSCLC patients receiving first-line target therapies (TT). Nevertheless, mainly due to the limited sample size and the unbalanced distribution of first-line treatments (CT-treated group, *n* = 14; IT-treated group, *n* = 4; TT-treated group, *n* = 5), we did not document any statistically significant difference.

## 4. Discussion

The present retrospective multicenter Italian study reports clinical outcomes of metastatic NSCLC patients harboring *BRAF* mutations. To date, despite available evidence in this setting of patients, data demonstrated is largely inconclusive, likely due to the relatively limited sample size and the retrospective nature of the studies. Thus, we sought to delve into this subset of lung cancer patients in an attempt to provide a real-life portrayal of the *BRAF* mutant population, particularly highlighting similarities and discrepancies between V600E and non-V600E groups.

The overall frequency of *BRAF* mutations was 5.4%, slightly higher than expected [10], but in line with a recent Italian survey [24]. At data cut-off, 44 consecutive *BRAF* mutant advanced NSCLC patients were included and the proportion of cases harboring *BRAF* V600E alteration resulted in 52.2%, consistent with other reported observations [10,11,12,21].

In addition to V600E, we explored the frequencies of rarer *BRAF* mutations, such as G469A/V/S, G466A, D594G/N, G464G, and Q257K.

When we analyzed the distribution of clinicopathologic features according to *BRAF* mutational subtypes (V600E vs. non-V600E), we observed that V600E mutations preferentially occurred in never-/light-smokers. This finding is in line with a previous meta-analysis, including 10 studies, in which no significant differences in *BRAF* mutation frequency were found in former/current smokers versus never smokers (OR = 0.95, 95% CI: 0.45–2.02), whereas the incidence of *BRAF* V600E significantly differed in former or current smokers versus never smokers (OR = 0.14, 95% CI: 0.05–042) [25]. No other relevant clinical profiles emerged to be associated with *BRAF* mutation subtype in our study. Specifically, we did not document any associations between gender, performance status, age, number, and location of metastatic sites, or *BRAF* mutational status. This evidence are similar to previously published works, confirming the smoking status as unique distinctive clinical characteristics of V600E mutant NSCLC [11,12,21].

The clinical outcome of patients with *BRAF* V600E mutation compared to those carrying non-V600E alterations was essentially comparable, with no significant differences in terms of survival and response to therapy. A favorable trend in PFS was observed in the V600E population, although without reaching statistical significance. This finding is partly in contrast with previous reports, demonstrating shorter PFS in V600E mutant patients, likely attributable to a more aggressive histotype, characterized by micropapillary features, typical of the V600E subgroup [13,21,26]. Considering the lack of such detailed histological information in our study, a conclusive judgment is not conceivable and future research involving a larger cohort of cases are warranted.

When focusing on the clinical outcome according to type of *BRAF* mutation and type of therapy, we found that median OS in the overall population treated with first-line platinum-based chemotherapy was 17.6 months, in line with that reported by previous works [21,27].

A worse performance of first-line immunotherapy versus chemotherapy was observed in terms of OS, either in the overall population or in the *BRAF* V600E subset. This latter finding partly mimics the preliminary results of Tan et al., although acknowledging their limited sample size [27]. Conversely, it has been demonstrated that, compared to other NSCLC harboring oncogene driver alterations, *BRAF* mutant patients could benefit from immunotherapy. Dudnik et al. conducted a retrospective study on a cohort of 39 patients positive for *BRAF* alterations and treated with ICIs. The study revealed a benefit from immunotherapy administered in first or subsequent lines, with an objective response rate (ORR) of 25–30% and a median PFS of 3.7–4.1, which are comparable to the results observed in the second-line setting in the unselected population of NSCLC [14]. No difference was observed in ORR or PFS according to the type of *BRAF* mutation. Similarly, a better outcome compared to other oncogene alteration was reported in terms of RR and PFS in the Immunotarget trial [13]. Finally, Rihawi et al. carried out a retrospective study on an Italian expanded access program comprising patients with advanced non-squamous NSCLC treated with second-line nivolumab [28]. The median OS were 10.3 and 11.2 months in *BRAF* mutated and wild type patients, respectively. The above-mentioned data demonstrates that immunotherapy could represent a reasonable treatment option after target therapy and chemotherapy in *BRAF* mutated NSCLC. At present, there are no data from randomized controlled trials comparing different first-line therapeutic strategies in *BRAF* mutated NSCLC and updated ESMO guidelines, although, when designating dabrafenib plus trametinib as the preferred regimen in a *BRAF* setting, it did not mention chemo-immunotherapy combinations, thus leaving this critical issue partly unanswered [29]. A recent retrospective study presented at ESMO 2021 Annual Congress using a propensity score weighting method showed that the first-line target therapy with dabrafenib plus trametinib reported a numerically longer OS and PFS over single-agent immunotherapy and chemo-immunotherapy [30]. In this regard, we did not detect any significant difference according to first-line regimens, likely depending on the relatively small sample size and the unbalanced distribution of first-line treatments among NSCLC patients.

Finally, we explored the association between *BRAF* variants and expression of PD-L1. A trend towards lower levels of PD-L1 was observed in the V600E population compared to the non-V600E subgroup. This result is inconsistent with previous evidence reporting high level of PD-L1 expression (TPS ≥ 50%) in the *BRAF* mutant population, with a similar proportion between V600E and the non-V600E subgroup and could partly explain the poor outcome obtained with immunotherapy [14].

Although the present multicenter Italian study addressed the clinically relevant goal to provide a real-life portrait of a still partially explored subset of advanced NSCLC carrying *BRAF* mutation, some limitations have to be acknowledged. First, the retrospective and non-interventional nature of our research, as well as the relatively limited sample size, prevents any conclusive judgement on different *BRAF*-driven subgroups. Second, the absence of a control group includes *BRAF* wild-type patients. Nonetheless, consistent with the literature, these fundamental unsolved issues require large prospective studies, involving *BRAF* mutant and wild-type cases, to be fully addressed.

## 5. Conclusions

This is one of the largest Italian series of advanced NSCLC patients with *BRAF* mutation, highlighting distinctive features of *BRAF* V600E and non-V600E mutant NSCLC in terms of survival outcome, therapeutic response, and clinicopathologic correlations. In particular, our study suggests that immunotherapy could not be the best option in *BRAF* mutant NSCLC patients, especially in V600E subtype, underlining the opportunity to consider a targeted therapy approach. To completely dissect the impact of target therapies, first-line chemotherapy- or immunotherapy-based treatment strategies, larger prospective studies are warranted.

## Figures and Tables

**Figure 1 cancers-14-02019-f001:**
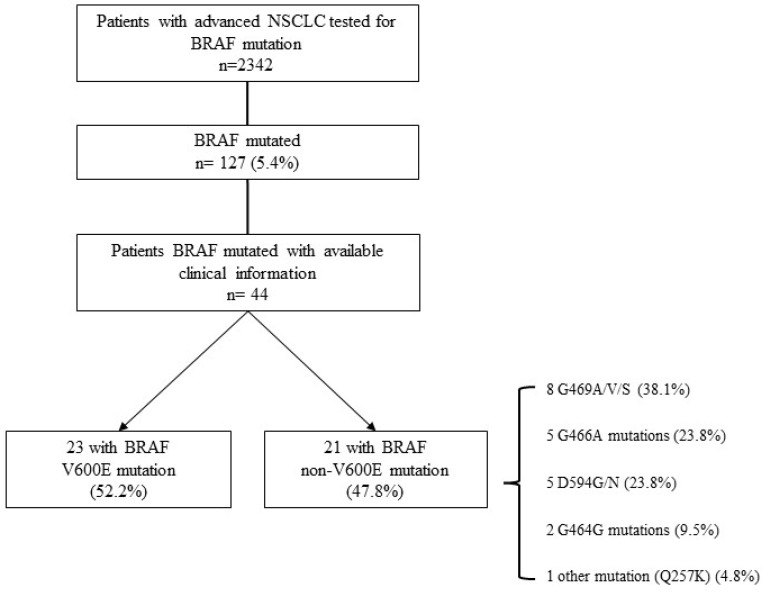
Consort diagram of DETECTION_KB study population: Advanced NSCLC patients included in this analysis, from 1 January 2016 to 28 February 2020.

**Figure 2 cancers-14-02019-f002:**
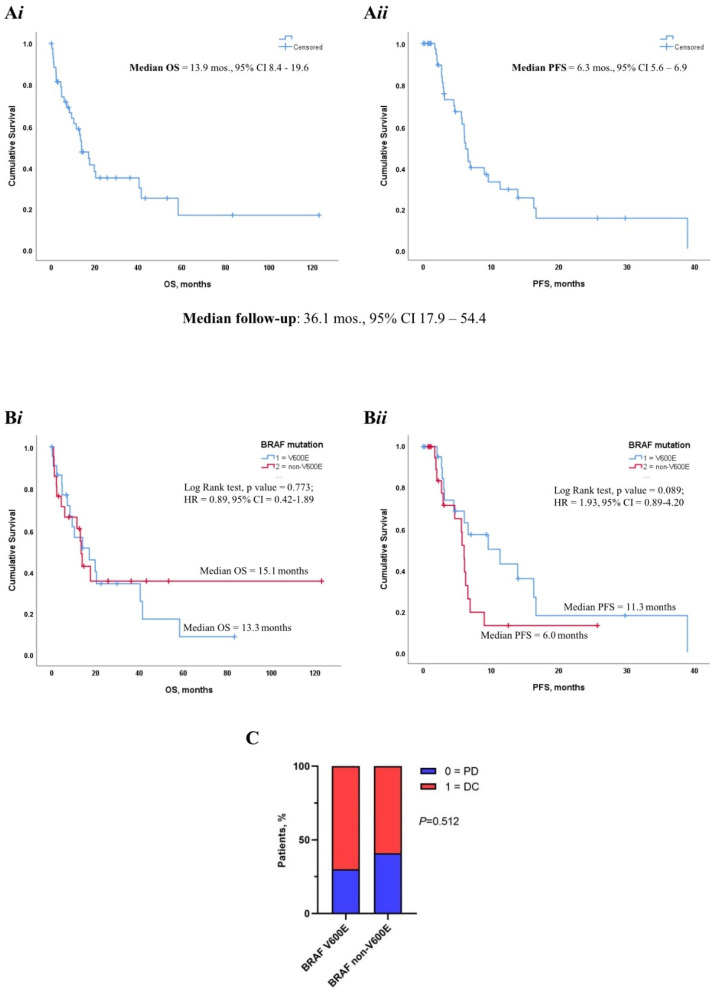
(**A*i***,**A*ii***) Median overall survival (OS) and progression-free survival (PFS) in the overall population. (**B*i***,**B*ii***) Kaplan Meier survival curves illustrating the survival outcome according to *BRAF* mutational status (V600E vs. non-V600E). (**C**) Stacked charts showing the treatment response (DC vs. PD) in *BRAF* V600E vs. non-V600E subsets. DC = CR + PR + SD ≥ 6 months; PD = PD + SD < 6 months.

**Figure 3 cancers-14-02019-f003:**
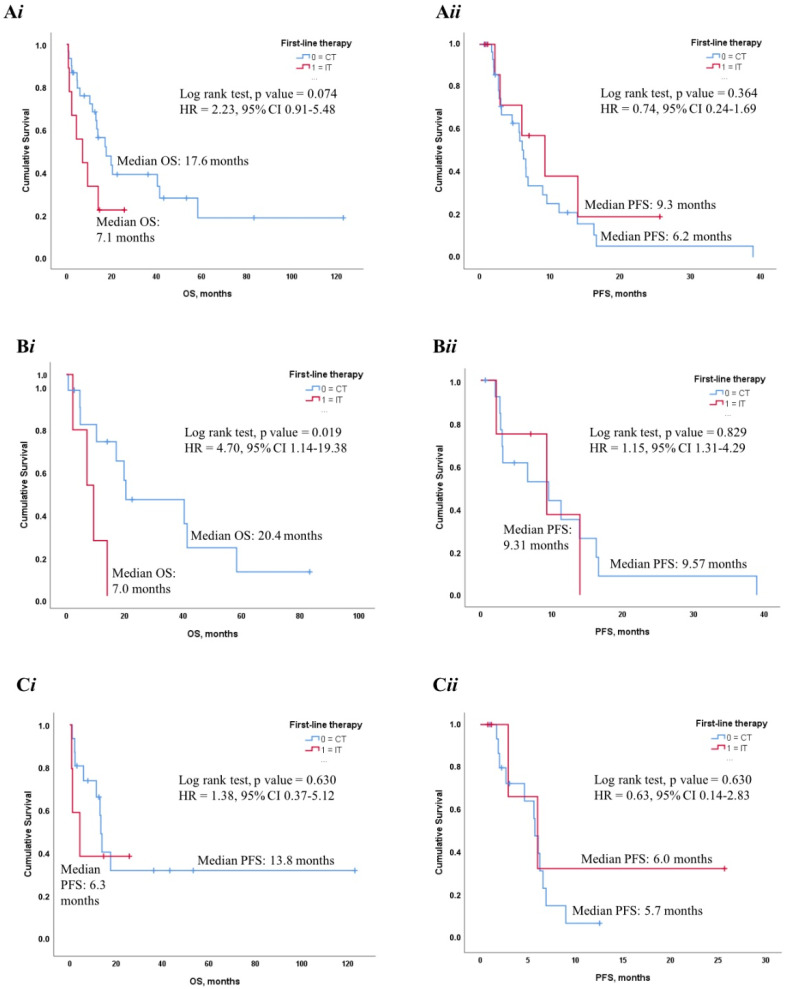
(**A*i***,**A*ii***): Kaplan Meier survival curves reporting the impact of first-line treatment (CT vs. IT) on OS and PFS in the overall population. (**B*i***,**B*ii***): Kaplan Meier survival curves reporting the impact of first-line treatment (CT vs. IT) on OS and PFS in the *BRAF* V600E group. (**C*i***,**C*ii***): Kaplan Meier survival curves reporting the impact of first-line treatment (CT vs. IT) on OS and PFS in *BRAF* non-V600E group. Abbreviations: CT, chemotherapy; IT, immunotherapy.

**Table 1 cancers-14-02019-t001:** Patient Population Characteristics. SCC: Squamous Cell Carcinoma; ADC: Adenocarcinoma; NSCLC: Non-Small Cell Lung Cancer; NOS: Not Otherwise Specified, ECOG: Eastern Cooperative Group; TPS: Tumor Proportional Score.

Patient Characteristics		Overall (*n* = 44)	*BRAF* V600E (*n* = 23)	*BRAF* Non-V600E (*n* = 21)	*p*-Value
Age, years (Median, range)		67 (47–84)	79 (47–81)	65 (49–84)	0.459
		*n* (%)			
Histotype	SCC	1 (2)	1 (4)	0 (0)	**0.024**
	ADC	36 (82)	17 (74)	19 (91)	
	NSCLC NOS	7 (16)	5 (22)	2 (9)	
Sex	Male	28 (64)	16 (70)	12 (57)	0.533
	Female	16 (36)	7 (30)	9 (43)	
Smoking status *	Smokers	9 (24)	3 (14)	6 (37)	**0.032**
	Ex-Smokers	23 (60)	13 (59)	10 (63)	
	Non Smokers	6 (16)	6 (27)	0 (0)	
ECOG PS	0–1	39 (89)	21 (91)	18 (86)	0.658
	2	5 (11)	2 (9)	3 (14)	
Stage of the disease	IIIB	4 (9)	2 (9)	2 (9)	0.946
	IV	40 (91)	21 (91)	19 (91)	
Number of metastatic sites	<3	17 (39)	10 (43)	7 (33)	0.548
	≥3	27 (61)	13 (57)	14 (67)	
Metastatic Involvement	Bone	14 (32)	7 (30)	7 (33)	1.000
	Brain	8 (18)	3 (13)	5 (24)	0.448
	Liver	7 (16)	5 (22)	2 (9)	0.416
	Lung	20 (45)	11 (48)	9 (43)	0.771
	Lymph nodes	32 (73)	15 (65)	17 (81)	0.318
	Adrenal gland	5 (11)	4 (17)	1 (5)	0.348
	Pleura	14 (32)	9 (39)	5 (24)	0.342
First-line Treatment	Chemotherapy	30 (68)	14 (61)	16 (76)	0.081
	Immunotherapy	9 (21)	4 (17)	5 (24)	
	Targeted therapy ^§^	5 (11)	5 (22)	0 (0)	
PD-L1 status	TPS ≥ 50%	16 (37)	30 (23)	47 (28)	0.513
	1% ≤ TPS < 50%	7 (16)	42 (32)	59 (35)	
	TPS < 1%	8 (18)	51 (39)	48 (29)	
	Unknown	13 (29)	7 (6)	13 (8)	

* In six patients, the data was not available. ^§^ Four patients were treated with Dabrafenib + Trametinib, one patient with Encorafenib + Binimetinib. *p* values are referred to the comparison between *BRAF* V600E and non-V600E groups (Fisher’s exact test for categorical variables; Mann Whitney test for continuous variables). Significant *p* values were reported in bold.

**Table 2 cancers-14-02019-t002:** Explanatory prognostic factors in Cox proportional hazard models: uni- and multivariate analysis illustrating the impact on OS and PFS of clinico-pathological characteristics.

OS, Univariate Analysis ^a^	Overall				*BRAF* V600E				*BRAF* Non-V600E			
	HR	CI (95%)	χ^2^	*p* Value	HR	CI (95%)	χ^2^	*p* Value	HR	CI (95%)	χ^2^	*p* Value
Age	1.05	1.01–1.10	5.73	**0.023**	1.02	0.96–1.08	0.54	0.477	1.07	1.01–1.15	4.75	**0.030**
Histotype	1.18	0.78–1.79	0.58	0.420	2.16	1.04–4.49	3.97	**0.040**	0.88	0.45–1.75	0.14	0.725
Sex	0.51	0.22–1.21	2.40	0.110	0.48	1.14–1.70	1.49	0.255	0.61	1.18–2.01	0.69	0.414
Smoking status	1.06	0.53–2.10	0.03	0.869	1.32	0.54–3.23	0.38	0.539	1.24	0.29–5.29	0.08	0.772
ECOG PS	4.17	1.38–12.63	4.77	**0.011**	1.39	0.18–11.04	0.09	0.754	18.6	3.0–75.68	8.89	**0.002**
N of metastatic sites	0.98	0.46–2.08	0.03	0.956	1.18	0.43–3.30	0.74	1.185	0.88	0.27–2.94	0.04	0.841
Bone metastasis	1.48	0.64–3.39	0.82	0.356	1.91	0.55–6.63	0.97	0.307	1.18	0.37–3.78	0.08	0.775
Brain metastasis	0.94	0.35–2.49	0.02	0.901	4.67	1.14–19.07	3.71	**0.032**	0.52	0.11–2.43	0.79	0.406
Liver metastasis	2.26	0.87–5.91	2.45	0.095	2.18	0.62–7.60	1.37	0.222	2.68	0.54–13.1	1.21	0.229
Lymphnodes metastasis	1.19	0.53–2.67	0.18	0.668	2.09	0.70–6.25	1.88	0.184	0.58	1.15–2.14	0.62	0.410
Adrenal metastasis	1.48	0.43–5.08	0.36	0.531	2.89	0.71–11.79	1.91	0.139	0.04	0.03–9.11	1.64	0.537
I line treatment	1.13	0.82–1.55	0.49	0.458	1.14	0.80–1.61	0.48	0.470	1.38	0.37–1.12	0.22	0.631
PD-L1 status, TPS	1.27	0.72–2.23	0.71	0.408	1.16	0.48–2.79	0.11	0.742	1.29	0.57–2.96	0.38	0.541
**PFS,** **Univariate Analysis ^a^**	**Overall**				***BRAF* V600E**				***BRAF* Non-V600E**			
	**HR**	**CI (95%)**	**χ^2^**	***p* Value**	**HR**	**CI (95%)**	**χ^2^**	***p* Value**	**HR**	**CI (95%)**	**χ^2^**	***p* Value**
Age	0.98	0.94–1.02	1.25	0.257	0.98	0.93–1.03	0.61	0.430	0.98	0.91–1.06	0.18	0.673
Histotype	0.83	0.52–1.34	0.65	0.448	1.11	0.52–2.38	0.08	0.780	0.76	0.38–1.51	0.75	0.440
Sex	0.91	0.42–1.97	0.06	0.810	1.47	0.48–4.54	0.43	0.504	0.19	0.05–0.79	5.97	**0.022**
Smoking status	1.44	0.77–2.68	1.32	0.254	0.89	0.37–2.17	0.06	0.805	1.61	0.54–4.82	0.71	0.393
ECOG PS	3.56	0.42–29.84	1.02	0.242	8.64	0.78–15.8	2.21	0.079	0.04	0.01–7.89	0.35	0.776
N of metastatic sites	1.39	0.62–3.11	0.66	0.422	1.53	0.49–4.72	0.57	0.455	0.87	0.27–2.81	0.05	0.813
Bone metastasis	2.60	1.16–5.81	5.09	**0.020**	2.22	0.64–7.69	1.45	0.209	2.60	0.89–7.56	2.99	0.079
Brain metastasis	2.31	0.84–6.37	2.28	0.103	2.28	0.47–11.1	0.89	0.306	3.63	0.79–16.5	2.48	0.096
Liver metastasis	1.19	0.47–2.99	0.13	0.714	1.75	0.51–6.03	0.73	0.378	0.99	0.21–4.54	0.01	0.986
Lymphnodes metastasis	0.96	0.43–2.15	0.01	0.925	0.99	0.33–2.99	0.01	0.993	2.17	0.04–2.78	4.35	**0.023**
Adrenal metastasis	1.60	0.54–4.75	0.65	0.395	0.48	0.61–9.01	1.43	0.204	1.37	0.17–11.09	0.08	0.766
I line treatment	0.79	0.51–1.21	1.48	0.278	0.87	0.56–1.33	0.48	0.512	0.63	0.14–2.82	0.41	0.545
PD-L1 status, TPS	0.72	0.40–1.28	1.21	0.267	0.69	0.26–1.83	0.53	0.455	0.96	0.44–2.09	0.01	0.910

PFS: progression free survival; OS: overall survival; Age (continue variable), Histotype (SCC = 0, ADC = 1, NOS = 3), Smoking status (negative smoking history = 0, positive smoking history = 1), Sex (Male = 0, Female = 1), Eastern Cooperative Oncology Group performance status (ECOG PS, 0–1 vs. 2), number of metastatic sites (<3 = 0, ≥3 = 1), bone/brain/liver/lymphnode/adrenal gland metastasis (absence = 0, presence = 1), first-line treatment (chemotherapy = 0, immunotherapy = 1, chemo-immunotherapy = 2), PD-L1 status (TPS < 1% = 0, 1–49% = 1, ≥50% = 2); Statistical results with *p* < 0.05 are bolded. ^a^ Univariate analysis is carried out without any adjustment.

## Data Availability

The data presented in this study are available on request from the corresponding author. The data are not publicly available in order to protect patients privacy.

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
