# Peer review of "Multicenter Observational Study on Metastatic Non-Small Cell Lung Cancer Harboring BRAF Mutations: Focus on Clinical Characteristics and Treatment Outcome of V600E and Non-V600E Subgroups"

_cancers, 2022, doi:10.3390/cancers14082019_

Round 1
Reviewer 1 Report
The aim of the study is clear. The methods are appropriate. The results are well summarized. The informativeness and style of the manuscript are excellent.
Minor concerns:
- The conclusion should be more specific. Not just an indication of what has been done in the study.
- The limitations of the work should be supplemented by indicating that there was a relatively small number of subjects to obtain some results and draw more conclusions.
Author Response
Reviewer #1:
The aim of the study is clear. The methods are appropriate. The results are well summarized. The informativeness and style of the manuscript are excellent.
We are grateful to the reviewer for these positive comments.
Minor concerns:
- The conclusion should be more specific. Not just an indication of what has been done in the study. We agree with these observations. The conclusion has been modified accordingly.
- The limitations of the work should be supplemented by indicating that there was a relatively small number of subjects to obtain some results and draw more conclusions. We are grateful to the reviewers for the observation. We have integrated the limitation of the study in the text.
Reviewer 2 Report
Thank you for giving me the opportunity to review this article.
This is a retrospective observational cohort study of BRAF mutation and lung cancer patients to describe their clinical course and prognosis.
Although the aim of the study seems achievable, the execution of this work could be improved:
The statistics need to be better specified. For table 1, please specify when the Fisher test or the X2 test is used.
Concerning the statistics, a lot of statistical tests are performed in table 1 and table 2. By performing a lot of tests, the authors increase the risk of finding a correlation simply because of the hazard. It is imperative that the authors apply corrections such as Bonferonni's to make the statistics presented acceptable.
I will remove the term "real life" from the title of the article. No study is truly a real life study and each has its own biases.
The title as well as the purpose of the study could be changed to "comparison of V600E and non-V600E mutations" which represent the bulk of the patients included in this work and probably the most interesting part of this work.
I did not detect any plagiarism or falsification of data. In total, I recommend acceptance of this manuscript after minor modifications.
Author Response
Reviewer #2:
This is a retrospective observational cohort study of BRAF mutation and lung cancer patients to describe their clinical course and prognosis.
Although the aim of the study seems achievable, the execution of this work could be improved:
The statistics need to be better specified. For table 1, please specify when the Fisher test or the X2 test is used. Thanks for the observation. For categorical variables, all the comparison between different groups have been computed by applying Fisher’s exact test, that is more appropriate for small-sized samples; for continuous variables Mann-Whitney test has been applied, in light of the not normal distribution of our data.
The statistical procedures employed in our study are specified in the lagend of Table 1 and detailed in the section 2.3 (statistical analysis) of Material and Methods.
Concerning the statistics, a lot of statistical tests are performed in table 1 and table 2. By performing a lot of tests, the authors increase the risk of finding a correlation simply because of the hazard. It is imperative that the authors apply corrections such as Bonferonni's to make the statistics presented acceptable. We fully agree with the raised issue, and we thank the reviewer for giving us the opportunity to implement the scientific and clinical relevance of our study. According to reviewer suggestion, Bonferroni’s correction test has been applied to all our multiple comparisons in order to minimize the bias related to the Multiplicity. Adjusted P-values are now reported in separate tables, provided as Supplementary Materials. We have also added specific sentences to Result section, detailing the outcome of this additional analysis. Specifically, for the multiple comparisons reported in Table 1, aimed at assessing potential differences in the distribution of clinico-pathological characteristics among BRAF V600E and non-V600E subgroups, Bonferroni’s correction has been conducted by adjusting the previously obtained P value for the number of comparisons.
Regarding the Bonferroni test for survival analysis, we did not observe any change in our previously significant findings, since the comparison aimed at assessing the impact of clinico-pathological characteristics on OS and PFS was conducted within each subgroup of BRAF mutated patients, without comparing BRAF V600E vs non-V600E.Finally, as specified in the Table legends, we would like to highlight that when P-values, corrected for the number of comparisons, exceeds 1, we reported just the value of 1.
I will remove the term "real life" from the title of the article. No study is truly a real life study and each has its own biases. The title as well as the purpose of the study could be changed to "comparison of V600E and non-V600E mutations" which represent the bulk of the patients included in this work and probably the most interesting part of this work. We agree with reviewer comments. We modified the title according to reviewer’s suggestions.
I did not detect any plagiarism or falsification of data. In total, I recommend acceptance of this manuscript after minor modifications. We are grateful to the reviewer for these positive comments.